# Climate Change Impacts on Rainfed Maize Yields in Kansas: Statistical vs. Process-Based Models

**Meenakshi Rawat** [1] , **Vaishali Sharda** [1,*] , **Xiaomao Lin** [2] **and Kraig Roozeboom** [2]

1   Carl and Melinda Helwig Department of Biological and Agricultural Engineering, Kansas State University, Manhattan, KS 66506, USA; rmeenakshi@ksu.edu

2   Department of Agronomy, Kansas State University, Manhattan, KS 66506, USA; xlin@ksu.edu (X.L.); kraig@ksu.edu (K.R.)

*   Correspondence: vsharda@ksu.edu; Tel.: +1-(510)-305-4262

**Abstract:** The changing climate and the projected increase in the variability and frequency of extreme events make accurate predictions of crop yield critically important for addressing emerging challenges to food security. Accurate and timely crop yield predictions offer invaluable insights to agronomists, producers, and decision-makers. Even without considering climate change, several factors including the environment, management, genetics, and their complex interactions make such predictions formidably challenging. This study introduced a statistical-based multiple linear regression (MLR) model for the forecasting of rainfed maize yields in Kansas. The model's performance is assessed by comparing its predictions with those generated using the Decision Support System for Agrotechnology Transfer (DSSAT), a process-based model. This evaluated the impact of synthetic climate change scenarios of 1 and 2 °C temperature rises on maize yield predictions. For analysis, 40 years of historic weather, soil, and crop management data were collected and converted to model-compatible formats to simulate and compare maize yield using both models. The MLR model's predicted yields (r = 0.93) had a stronger association with observed yields than the DSSAT's simulated yields (r = 0.70). A climate change impact analysis showed that the DSSAT predicted an 8.7% reduction in rainfed maize yield for a 1 °C temperature rise and an 18.3% reduction for a 2 °C rise. The MLR model predicted a nearly 6% reduction in both scenarios. Due to the extreme heat effect, the predicted impacts under uniform climate change scenarios were considerably more severe for the process-based model than for the statistical-based model.

**Keywords:** climate change impacts; DSSAT; model inter-comparison; maize; multiple linear regression (MLR) model



## 1. Introduction

Globally, the average temperature and precipitation have increased at an average rate of 0.18 °C [1] since 1981 and 1.016 mm per decade since 1901 [2], respectively. For the United States, this average rate of increase in temperature and precipitation has been 0.27 °C since 1981 [3] and 5.08 mm per decade since 1901 [2], respectively. The state of Kansas forms a part of the central Great Plains and lies in the western boundary of the United States Corn Belt, and faces an east-to-west precipitation gradient of climatology and its changes [4]. During 1895–2015, the average temperature in Kansas increased by 0.06 ± 0.03 °C per decade, with annual mean precipitation for the western third, the central third, and the eastern third of Kansas being 531 mm, 660 mm, and 945 mm, respectively [5]. These uneven changes in temperature from north to south, and in precipitation from east to west, have been shown to impact crop productivity [6].

Maize (*Zea mays* L.) is the major crop grown both in the USA and globally, and the USA contributes to 50% of global maize production [7,8]. It is a global staple food, a major source of the human diet as well as livestock feed, and has varied industrial and energy uses [9]. Maize production in Kansas ranks sixth in the nation and is the second most grown crop

in Kansas [10] after winter wheat in terms of bushels produced, with the Kansas maize industry playing a significant role in the US agricultural economy [11].

However, the changing climate adversely affects crop production [12], which is a challenge in the face of a growing global population. The changing climate results in frequent warming events and irregular precipitation [13], which in turn impact crop production and induce biotic [14] and abiotic stresses [15]. The impact of the climate on crop yield varies by crop type and geographical location [16,17], and precise and timely predictions of crop yield can provide valuable information to agronomists, producers, and decision-makers [18–21]. It has been shown that increasing temperatures reduce maize yields in the US Midwest, with higher temperatures being associated with short grain filling periods that hasten reproductive development [22]. Timely precipitation could mitigate rising temperatures, though significant yield losses have been reported due to fluctuations in precipitation [13]. It has been found that a 1 °C rise in the maximum daily temperature ($T_{max}$) has reduced maize and rice yields in the southeastern USA by 34% and 8.3%, respectively [23], and results from a statistical model have indicated an 8.3% maize yield reduction globally with a 1 °C rise in temperature [24]. A recent study [25] conducted in Kansas found that temperature has a more pronounced influence on maize production in the region than rainfall. Maize yield reductions in the US of 43% to 44% have been predicted under a slow-warming scenario (assuming resource-efficient technology) and 74% to 79% yield reductions in a fast-warming scenario (assuming the continued use of fossil fuels, which results in the largest increase in $CO_2$ concentrations and temperatures [26,27]). The changing climate and the projected increase in the variability and frequency of extreme events make accurate predictions of crop yield critically important for addressing emerging challenges to food security [10,28].

To assess the impact of climate change on crop yield, two distinct approaches have been used worldwide. The first approach uses process-based models that require detailed input data for soils, weather, and crop management practices to simulate in-season crop growth and end-of-season crop yield. The second approach links crop yield data to weather variables to make predictions using statistical methods. Both approaches have their strengths and weaknesses [10,29].

Though several studies have considered the impacts of future climate change on crop production [12,13,16,23,27,30–34], and have studied comparisons and combinations of process-based, statistical regression, and machine learning models [29,35,36], these models have not been compared for their yield prediction performance and the impact of climate change on yield at a county scale. Therefore, this study was undertaken to develop a new statistically based multiple linear regression (MLR) model for predicting rainfed maize yield at the county scale in Kansas; to compare the predicted yield from the MLR model with that of a process-based model (DSSAT); and to assess the performance of both models for predicting the impact of climate change on maize yield for synthetic climate change scenarios of 1 and 2 °C temperature rises [10] from 1981 to 2020.

## 2. Materials and Methods

### 2.1. Study Region

The selected study region (Figure 1) is the state of Kansas, located in climatic zones B and C, which comprise dry and moist subtropical mid-latitude climates [37]. Zone B experiences potential evaporation and transpiration rates exceeding precipitation, resulting in dry conditions, whereas Zone C has warm and humid summers with mild winters [37]. Kansas is a part of the midwestern United States and extends from 36°59′36.11″ N to 40°00′10.07″ N, and from 94°35′18.20″ W to 102°03′04.43″ W [10]. The selection of the 40 Kansas counties included in the analysis, as shown in Figure 1, was based on data availability for maize grown under rainfed conditions, which is discussed in further detail.

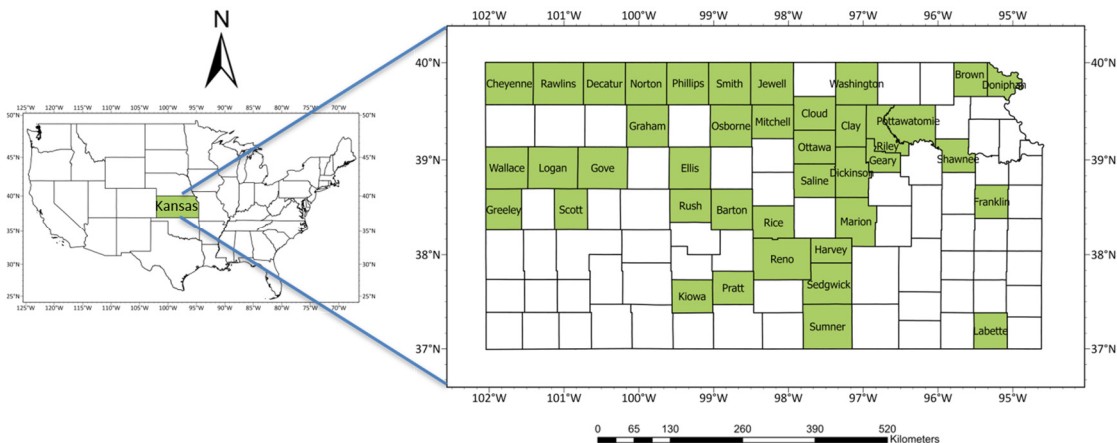

**Figure 1.** Map of the study area showing the 40 counties used for maize yield predictions from statistical and process-based models.

*2.2. Models and Models' Parameters*

2.2.1. Multiple Linear Regression (MLR) Model

The relationship between the time series of historic rainfed maize yield and climate variables was modeled with MLR at the county level. We assumed that residuals were normally distributed and there was no autocorrelation between predictors [38]. Candidate predictor variables for the MLR models were the growing degree days (GDD), extreme degree days (EDD), the total precipitation for the month of May (PRECIP-May), the total precipitation for the month of June (PRECIP-June), the total precipitation for the month of July (PRECIP-July), the total precipitation for the month of August (PRECIP-August), the average temperature for the month of May (TEMP-May), the average temperature for the month of June (TEMP-June), the average temperature for the month of July (TEMP-July), and their interactions [10].

The GDD are used to estimate the thermal time that is required for a specific cultivar/crop to develop or accumulate exposure to heat over the growing season.

$$GDD_d = \frac{T_{X,d} + T_{N,d}}{2} - T_{baseline},\tag{1}$$

$$\text{where, } T_{X,d} = \begin{cases} T_{X,d} & \text{if } T_{baseline} < T_{X,d} < T_{high}, \\ T_{baseline} & \text{if } T_{X,d} \leq T_{baseline}, \\ T_{high} & \text{if } T_{X,d} \geq T_{high}. \end{cases}$$

where $GDD_d$ (Equation (1)) [10] is the daily heat unit defined on each day, d, $T_{high}$ is the high temperature (typically set to 29 °C), $T_X$ is the maximum daily air temperature, $T_N$ is the minimum daily air temperature, and $T_{baseline}$ is the baseline temperature (typically set to 9 °C) [39,40].

The EDD accounts for growing degree days above the threshold, with a high of $\left(T_{high}\right)$ 29 °C for maize. It generally has a strong negative relation with maize yields [41], and is defined as

$$EDD_d = \begin{cases} T_{X,d} - T_{high} & \text{if } T_{X,d} > T_{high}, \\ 0 & \text{if } T_{X,d} \leq T_{high}. \end{cases}\tag{2}$$

where $EDD_d$ (Equation (2)) [10] is the daily heat unit above the threshold, $T_{high}$ (29 °C), defined on each day, d.

After obtaining the daily GDD and EDD from Equations (1) and (2), respectively, we calculated the accumulated GDD and EDD for the growing season of maize (April to September). From the time series data obtained from the USDA-NASS, we observed that maize yields generally increased with time due to technological advancement, genetic improvement, changes in management practices, etc. [42]. To isolate the

variability of rainfed maize yield due only to climate variables, the time series data were detrended linearly to remove the trend of technological advancement on yields and interannual variability within climate variables. Linear detrending is the most common detrending process and consists of removing a straight line trend component from a time series [43]. Detrending also minimizes the influence of slowly changing factors such as soil and crop management [24]. Therefore, all climate variables (predictors) and rainfed maize yields (response variable) were detrended linearly. Then, county-specific MLR models were developed to evaluate the relationship between detrended rainfed maize yield and detrended climate variables from 1981 to 2020. We developed MLR models for each county using a 5-fold cross-validation (5-fold CV) method [29]. This method divides the data into five roughly equal parts, then trains the model on four parts and tests the model on the fifth. This procedure was repeated for all five parts of the data, then the $R^2$ and RMSE were obtained for all tested parts and averaged at the county scale. Six variables out of the nine variables resulting in high $R^2$ and low RMSE for each county were chosen (4 independent and 2 interactions) to develop a model in the form given in Equation (3) [10]:

$$Y'_{maize} = \alpha_0 + \beta_1 P'_1 + \beta_2 P'_2 + \beta_3 P'_3 + \beta_4 P'_4 + \beta_5 P'_i P'_j + \beta_6 P'_k P'_1 \tag{3}$$

where $Y'_{maize}$ is the detrended predicted maize yield, $\alpha_0$ is the intercept, $\beta_1$–$\beta_6$ are the estimated coefficients of 4 independent variables and 2 interaction variables, $P'_1$–$P'_4$ are the detrended independent predictor variables, and $P'_I P'_j$–$P'_k P'_1$ are the detrended interaction predictor variables. Box plots were used to visualize the range and variability of county-level independent variable coefficients.

### 2.2.2. Process-Based Model, DSSAT

The process-based crop simulation model, the Decision Support System for Agrotechnology Transfer (DSSAT), version 4.8 [44], was used to simulate maize yield for the years 1981–2020. DSSAT version 4.8 includes models for 42 crops, and the model can simulate crop growth, development, and yield for user-defined management strategies [45]. DSSAT's Crop Estimation through Resources and Environmental Synthesis (CERES) Maize [46] model calculates maize growth and the end-of-season crop yield and yield components, along with the daily nutrient and water balance. The CERES Maize model simulates six different phenological stages for a maize plant, with each stage controlled by the genetic traits of the cultivar and their interactions with the environment [47]. For this study, we used maize cultivar PIO 3489, which has been calibrated for north-east Kansas [48]. The DSSAT requires daily weather data, soil information at a different depths of the soil profile, detailed crop management information, and cultivars as input parameters to simulate crop growth and development at a specific location [10,44].

After obtaining climate, rainfed maize yields, crop management, and soil properties datasets, the area under rainfed maize cultivation was extracted using information from the Global Map of Irrigation Areas (GMIA) [49] and CropScape [50]. The data from CropScape were available at a 30 m resolution, whereas the GMIA data for the agriculture area equipped for irrigation (AEI) were at a 10 km resolution. The GMIA data were downscaled to 4 km, with the rainfed grid being defined as having an AEI of less than 5%. The data were processed in ArcGIS Pro 3 [51] to obtain the rainfed maize yield gridded area for each county, and was then merged with the pre-processed gSSURGO soil data. The gridded soil data were converted into a DSSAT-compatible format, and rainfed maize yields were simulated at a 4 km grid using the DSSAT model. The model was run for 1114 rainfed grid points, with 702 dominant soils across 40 counties. Then, the gridded simulated yields were aggregated at the county scale and averaged over dominant soil types for 40 years [10]. The DSSAT simulated yields were also detrended linearly as discussed above in MLR for the comparison of predicted yields from the models.

### 2.3. Data and Data Sources

#### 2.3.1. Climate Data

The daily historic temperature and precipitation data that were used for this study were obtained from the Parameter-elevation Relationships on Independent Slopes Model (PRISM) [52] and include the daily maximum temperature ($T_X$), the daily minimum temperature ($T_N$), and precipitation (PRECIP) at a 4 km grid. The gridded raster data for these climate variables for the years 1981–2020 were processed and aggregated at the county level for each growing season [53]. Daily solar radiation data, available from NASA POWER [54] at a 1° resolution, were interpolated to county-level radiation data. Kansas has varying temperature and precipitation from county to county, with an east to west decreasing precipitation gradient with average annual precipitation ranging from 1172 mm to 435 mm, and a north to south increasing average annual temperature trend ranging between 10.7 °C and 14.3 °C (40 years average, Figure 2a,b [10,52]).

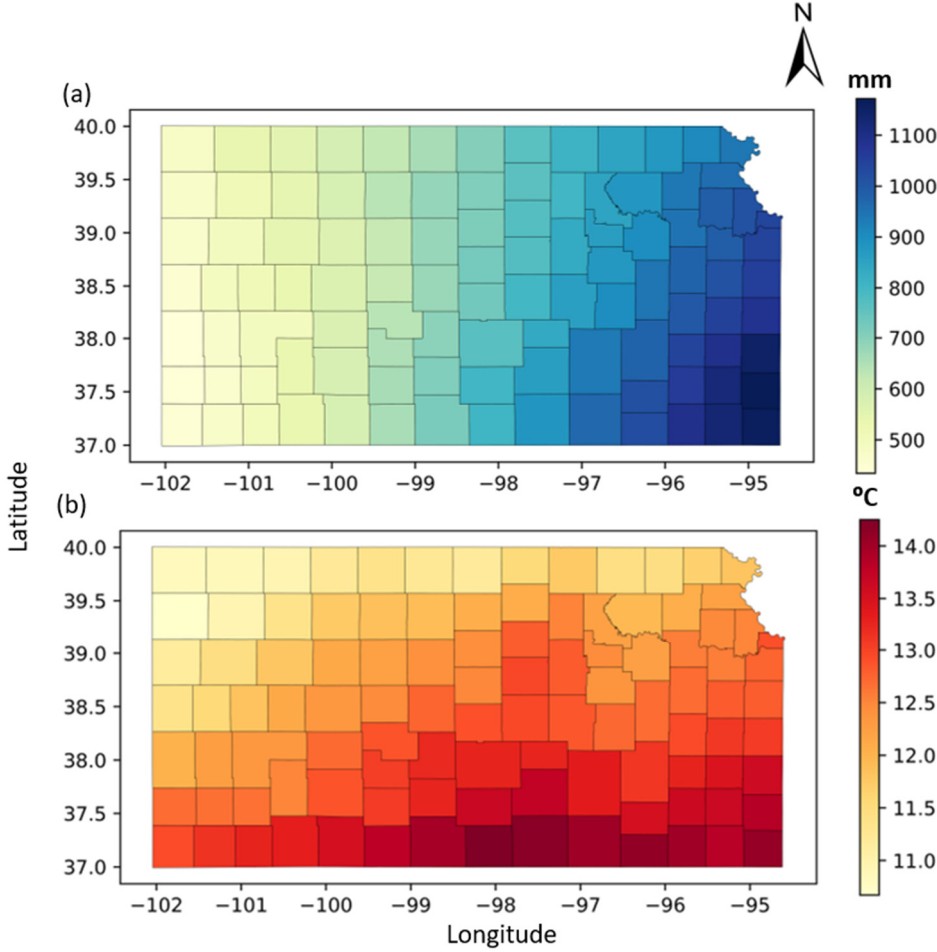

**Figure 2.** (**a**) Annual precipitation; (**b**) annual mean temperature across Kansas from 1981 to 2020.

#### 2.3.2. Soil Data

The soil data were retrieved from the Natural Resource Conservation Service (NRCS) Gridded Soil Survey Geographic (gSSURGO) database, available at a 30 m resolution. These data were aggregated to a 4 km grid scale, and ArcGIS 10.8 (Soil Data Development Toolbox) was used to extract the selected physical, chemical, and engineering properties for Kansas counties. These properties include the dry bulk density at 1/3 bar pressure, clay%, silt%, sand%, soil pH, saturated hydraulic conductivity, organic matter, available water content at 1/3rd bar and 15 bar pressure, cation exchange capacity, % slope, farmland classification, runoff, and drainage class [10].

### 2.3.3. Rainfed Maize Yield Data

Historical rainfed maize yield data for Kansas counties were obtained from the USDA-NASS Quick Stats database and K-State Research and Extension (KSRE) reports [55]. For this study, we used annual county-level rainfed maize grain yields from 1981 to 2020. We selected counties that had historic rainfed yield data available for at least 20 years within the period of study (1981–2020). This resulted in a selection of 40 counties out of the 105 counties in Kansas. Crop management data were obtained from the USDA-NASS and KSRE [55]. This included a plant population of 7.6 plant/$m^{-2}$, a row spacing of 0.51 m, and a total nitrogen fertilizer application in the form of urea at 170 kg/$ha^{-1}$. Urea application was divided into two equal halves, with the first half applied before planting and the second half during side-dressing and fertigation. To ensure the comparability of the results, we selected April to September as the growing season during the multiple linear regression (MLR) simulations consistently across all counties in Kansas. In contrast, the growing seasons varied from county to county in the DSSAT simulations due to variations in the planting dates and crop management [10].

### 2.4. Model Evaluation

To evaluate the performance of the MLR and DSSAT models, the predicted yields ($y_{pred}$) from both models were compared with the observed yield ($y_{obs}$) using statistical metrics [56]. These were the root mean square error (RMSE), Pearson correlation coefficient ($r$), and coefficient of determination ($R^2$) given in Equations (4), (5), and (6), respectively [10].

The RMSE measures the difference in magnitude between predicted and observed values and is given as

$$\text{RMSE} = \sqrt{\frac{1}{n}\sum_{i=1}^{n}\left(y_{obs,i} - y_{pred,i}\right)^2} \tag{4}$$

where $y_{obs,i}$ and $y_{pred,i}$ are the observed and predicted yield for the $i^{th}$ data record, respectively.

The Pearson correlation coefficient, r, measures the degree of linear association between predicted and observed values, and ranges between $-1$ to $+1$, where $\overline{y}_{obs}$ and $\overline{y}_{pred}$ are the average observed and predicted yield, respectively.

$$r = \frac{\sum\limits_{i=0}^{n}\left(y_{obs,i} - \overline{y}_{obs}\right)\left(y_{pred,i} - \overline{y}_{pred}\right)}{\sqrt{\sum\limits_{i=0}^{n}\left(y_{obs,i} - \overline{y}_{obs}\right)^2}\sqrt{\sum\limits_{i=0}^{n}\left(y_{pred,i} - \overline{y}_{pred}\right)^2}} \tag{5}$$

The coefficient of determination, $R^2$, measures how much variance in the observed yield can be explained by predictors in a linear regression fit, and is given as

$$R^2 = 1 - \frac{\sum\limits_{i}\left[y_{obs,i} - y_{pred,i}\right]^2}{\sum\limits_{i}\left(y_{obs,i} - \overline{y}_{obs}\right)^2} \tag{6}$$

### 2.5. Synthetic Climate Change Scenarios

To assess the impact of temperature rises on maize yield, as simulated with the MLR and the process-based CERES Maize models, we developed synthetic climate datasets to mimic the changes in climate that may occur in the future. For this, we chose scenarios of 1 and 2 °C rises in daily temperature. It is important to note that these scenarios are not tied to any future global circulation models and are hypothetical [29]. After adjusting the historical temperature dataset by increasing the daily temperatures by either 1 or 2 °C, both the MLR and the DSSAT models were run to simulate new maize yields under the two synthetic climate change scenarios. The differences were calculated between the new

predicted county-level yield for each scenario of 1 and 2 °C rises and the original predicted yield data. The changes in yields that were predicted using both models for each scenario were plotted spatially, and bar plots were developed to evaluate the overall yield changes across Kansas [10].

## 3. Results and Discussion

### 3.1. Multiple Linear Regression (MLR) Model

The $R^2$ values for the MLR models for the 40 counties of Kansas ranged from 0.32 to 0.87 (Figure 3) [10], which means that at least 32% of the variability in the year-to-year maize yield change was explained by climate variables, with maximum and minimum variability observed in Rice and Graham counties, respectively. The variability in the $R^2$ showed a similar pattern to the variability of the PRECIP-July across Kansas, providing evidence that much of the model's explanatory power was derived from the PRECIP-July. The lower $R^2$ that was observed in a few counties could be attributed to the high year-to-year fluctuation in rainfed maize yield in those counties. The wide range of $R^2$ values could likely be attributed to missing and non-uniform observed yield data, variations in the length of the growing seasons among counties, climate responses at the county scale, and changes in economics or other conditions that influence crop management. A better consideration of these factors would likely improve the performance of the model. However, the MLR model was able to explain 30% to 90% of the overall variability in the rainfed maize yield. Our county-scale results exhibit a range of values from being relatively lower to higher, and are comparable to global and state-scale studies that can be found in the literature. A study [24] found that a 47% year-to-year yield variability in maize yield at a global scale was explained by climate variables. Similarly, another study [41] reported an $R^2$ value of 0.72 for a regression model between maize yields and climate variables for Illinois.

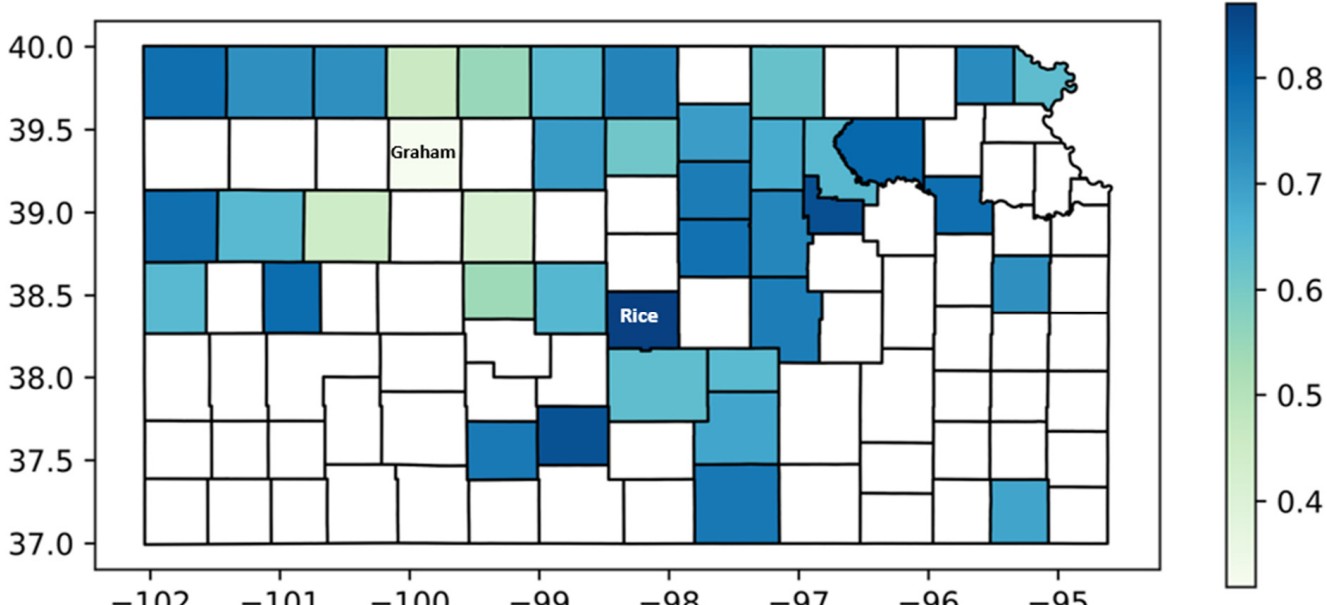

**Figure 3.** Spatial distribution of $R^2$, values obtained from the MLR model for selected rainfed maize-producing counties of Kansas, with Rice and Graham counties labeled.

The RMSE for the maize yield predictions ranged from 483 kg ha$^{-1}$ for Pratt County to 1388 kg ha$^{-1}$ for Ellis County (Figure 4) [10]. The likely cause of the wide range of RMSE values is due to the non-uniform spatial distribution of the rainfed maize yield data from 1981 to 2020. The ranges of the NASS rainfed maize yields from 1981 to 2020 across Kansas were 200–1200 kg ha$^{-1}$ in the north-east region, 100–700 kg ha$^{-1}$ in the north-west region, and 500–800 kg ha$^{-1}$ in the central region. The variability in these yields can be attributed

to the year-to-year variability in temperature and precipitation. In certain years, the rainfed yields were considerably less in the western and central regions due to drought and heat, particularly in 1983, 2000, 2011, and 2012. The models' RMSE results are similar to those reported for studies that have considered soil parameters in addition to climate variables in their statistical models to predict maize yield [28], as well as to studies conducted using a suite of statistical and machine learning models for maize [57] and soybean [58].

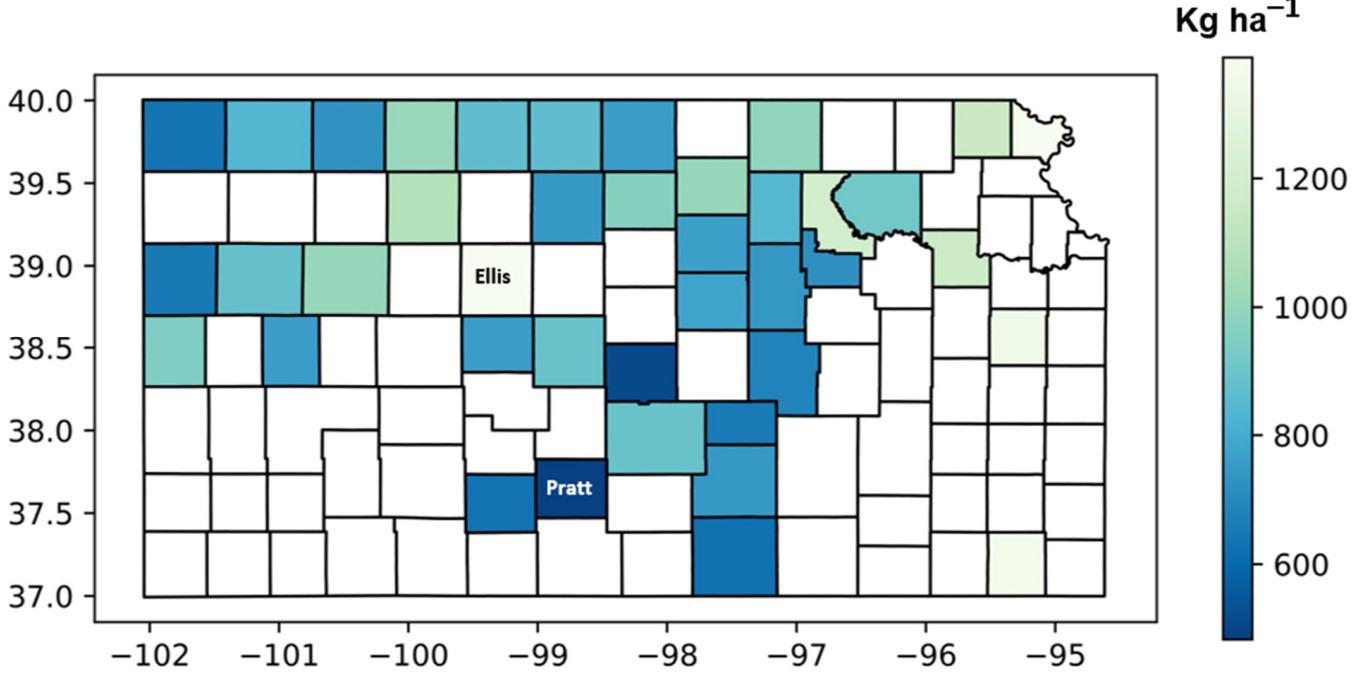

**Figure 4.** Spatial distribution of RMSE values obtained from the MLR model for selected rainfed maize-producing counties of Kansas, with Ellis and Pratt Counties labeled.

Figure 5 [10] shows the range of coefficients of the predictor variables for the county-level MLR models, with counts representing the number of counties having those variables in their MLR model. For the MLR models, 60% of the counties had GDD and 70% had EDD, followed by 55% with PRECIP-June and 52% with PRECIP-July as predictor variables. The predictor variables with predominantly positive relationships with the maize yield were GDD, PRECIP-July, and TEMP-May * $10^{-1}$ where * represents multiplication. The predictor variables with generally negative or strongly negative relationships with maize yield were EDD, PRECIP-May, PRECIP-Aug, TEMP-June * $10^{-1}$, and TEMP-July * $10^{-1}$. A study [41] also found that GDD and EDD showed a positive and negative relationship with the maize yield, respectively [10].

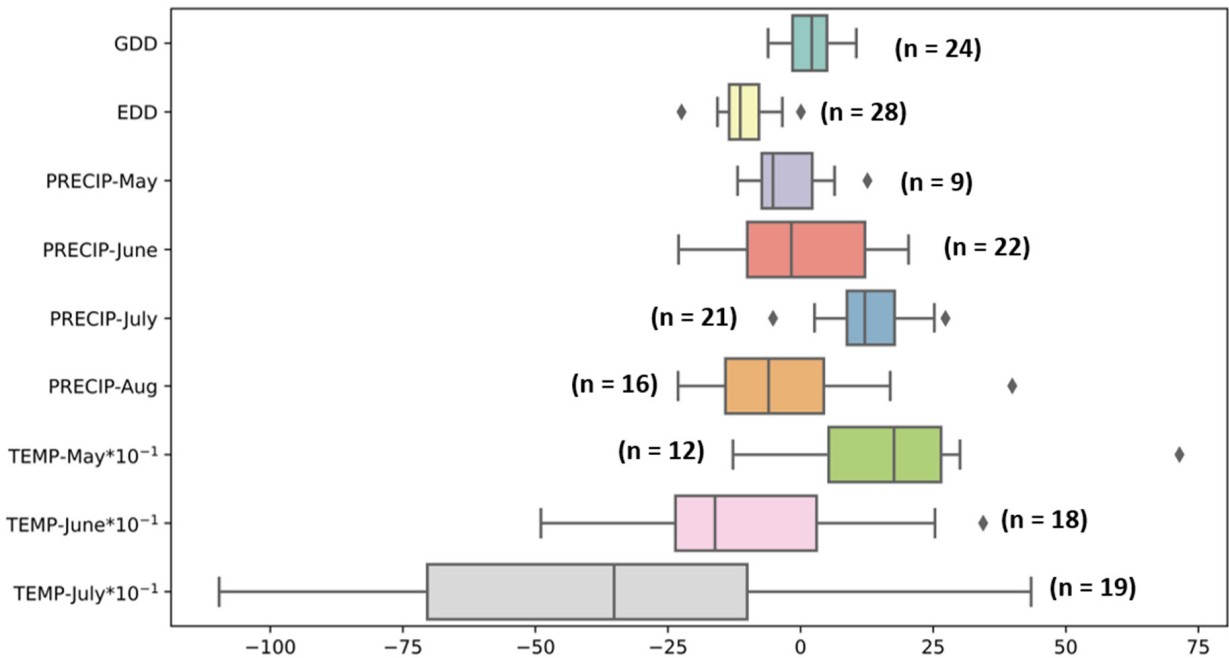

**Figure 5.** Range of coefficients for variables of county-level MLR models, with counts for selected rainfed maize-producing counties of Kansas. The coefficients are the growing degree days (GDD), extreme degree days (EDD), total precipitation for the month of May (PRECIP-May), total precipitation for the month of June (PRECIP-June), total precipitation for the month of July (PRECIP-July), total precipitation for the month of August (PRECIP-August), average temperature for the month of May (TEMP-May), average temperature for the month of June (TEMP-June), and average temperature for the month of July (TEMP-July) where * denotes multiplication.

July's precipitation and temperature are important parameters for predicting maize yield because the critical kernel set and early grain fill stages typically occur in this month, and are both sensitive to moisture and temperature extremes [22]. During flowering, high daytime temperatures can result in pollen sterility, and reduced seed count, whereas high night-time temperatures result in a faster growing degree day accumulation that often leads to a shorter grain fill duration and early maize maturation [59–61]. The non-uniform distribution of these variables of the county-level MLR across Kansas depends upon the location of the county, weather conditions, and management practices. Our analysis revealed that the GDD emerged as a crucial predictor in the north-eastern and central regions of Kansas, but the EDD and PRECIP-July did not exhibit any specific pattern across the state. Understanding the dynamics of these variables is important for making informed decisions about crop growth and development, particularly in regions where farmers rely heavily on precipitation for agricultural production. By having accurate knowledge of local climate conditions, implementing appropriate management practices, and understanding the specific requirements of crop growth in these areas, we can strive towards achieving improved yields that directly benefit local farmers. This insight can pave the way for more sustainable and efficient agricultural practices, enhancing the overall resilience of the farming communities in Kansas [62].

### 3.2. Comparison of Regression and Process-Based Models

The overall performance of both models was assessed using statistical metrics, and time series were developed at the county level. Due to the challenges associated with evaluating the statistical metrics for each county, we evaluated the performance across the state. A statewide approach enabled a broader evaluation of the models' performance in capturing the overall patterns and trends in maize yield across Kansas. The MLR model achieved an r value of 0.93 and an RMSE of 443 kg ha$^{-1}$, while the DSSAT model achieved

an r value of 0.70 and an RMSE of 1408 kg ha$^{-1}$ [10]. The MLR model was able to capture a significant portion of the variability in the observed NASS yields, implying a good fit to the data, and its lower RMSE indicates good accuracy in predicting maize yields in Kansas. After performance evaluation, time series were developed for the detrended predicted yields from both models, with an observed detrended yield across Kansas as shown in Figure 6 (see Supplementary Material, Figure S1 to access the time series for each Kansas county).

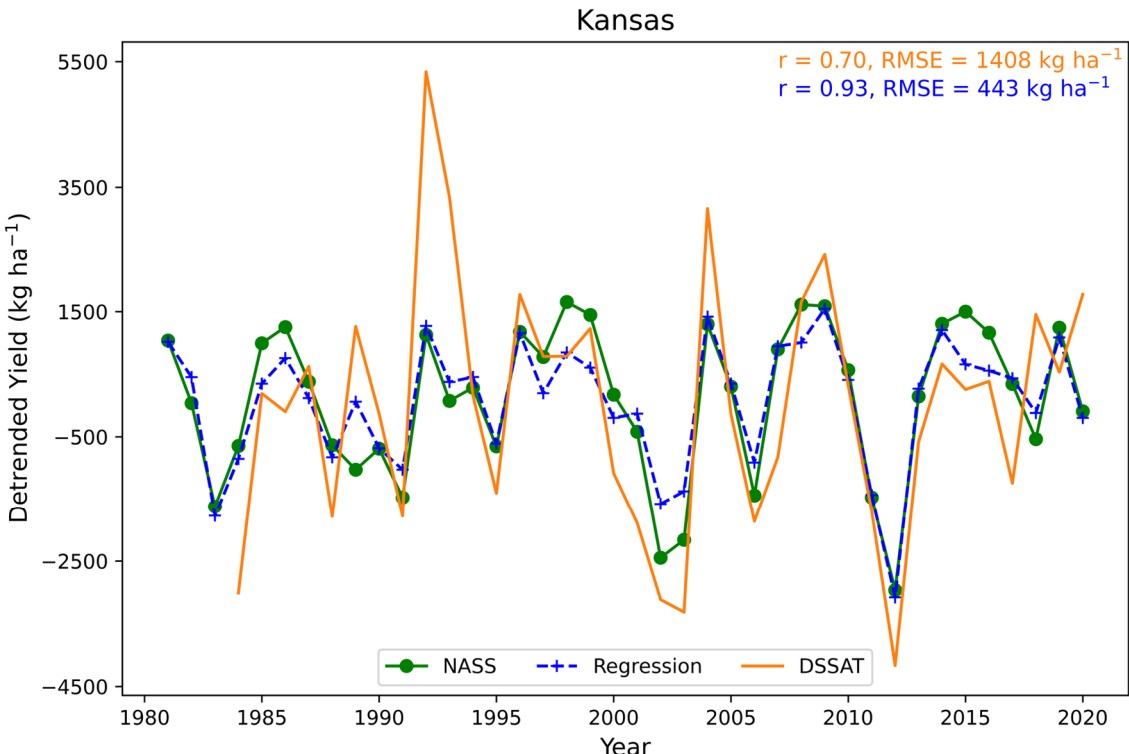

**Figure 6.** Time series plot of historic NASS yields (observed) and predicted detrended yields from the MLR and DSSAT models for Kansas.

A study by Liu et al. [63] also compared rainfed maize yield from multiple machine learning (ML) and process-based DSSAT models for the US Corn Belt, and found that the RMSE was about three times higher for DSSAT yield predictions compared to the best performing ML model, and DSSAT yield estimates explained 16% of the spatiotemporal variance of the observed maize yield. We compared the results of our study with the previous studies to show how our findings align with or differ from other studies [64,65]. It is important to exercise caution when interpreting the results of the comparisons obtained in our study due to the differences in calibration processes. While the MLR model was trained on observed yield data from 1981 to 2020, limited calibration was performed on the DSSAT model [36]. Also, the DSSAT yields were simulated for 1984 to 2020, whereas the MLR-simulated yields were based on observed (USDA-NASS) maize yield data availability. Nevertheless, it is essential to highlight the promise of the MLR model for yield prediction compared to process-based models because it requires considerably fewer inputs and produces time series patterns that are similar to the observed yield [10].

### 3.3. Climate Change Impacts on Predicted Yields

The comparison of the predicted maize yield from both models for two synthetic climate change scenarios of 1 and 2 °C rises in temperature (Scenarios I and II, respectively) showed that reduction in the predicted maize yield is greater in the DSSAT model compared to the MLR model (Figures 7 and 8) [10]. Using the MLR model, seven of the forty counties

that were included in the analysis showed an increase in predicted maize yield for both scenarios. In Scenario I, the predicted maize yield from the MLR model showed increases in the range of 20 to 469 kg ha$^{-1}$, and decreases in the range of 19 to 1118 kg ha$^{-1}$. In Scenario II, the range of increases in predicted yield varied between 8 and 455 kg ha$^{-1}$, and the decreases ranged from 19 to 1136 kg ha$^{-1}$. The DSSAT showed reductions in the predicted rainfed maize yield for both scenarios. In Scenario I, reductions in the predicted maize yield ranged between 280 and 980 kg ha$^{-1}$, whereas, in Scenario II, reductions ranged from 590 to 1810 kg ha$^{-1}$ [10]. The differences in yield reductions under both scenarios, as predicted using the MLR and DSSAT models, could have several causes.

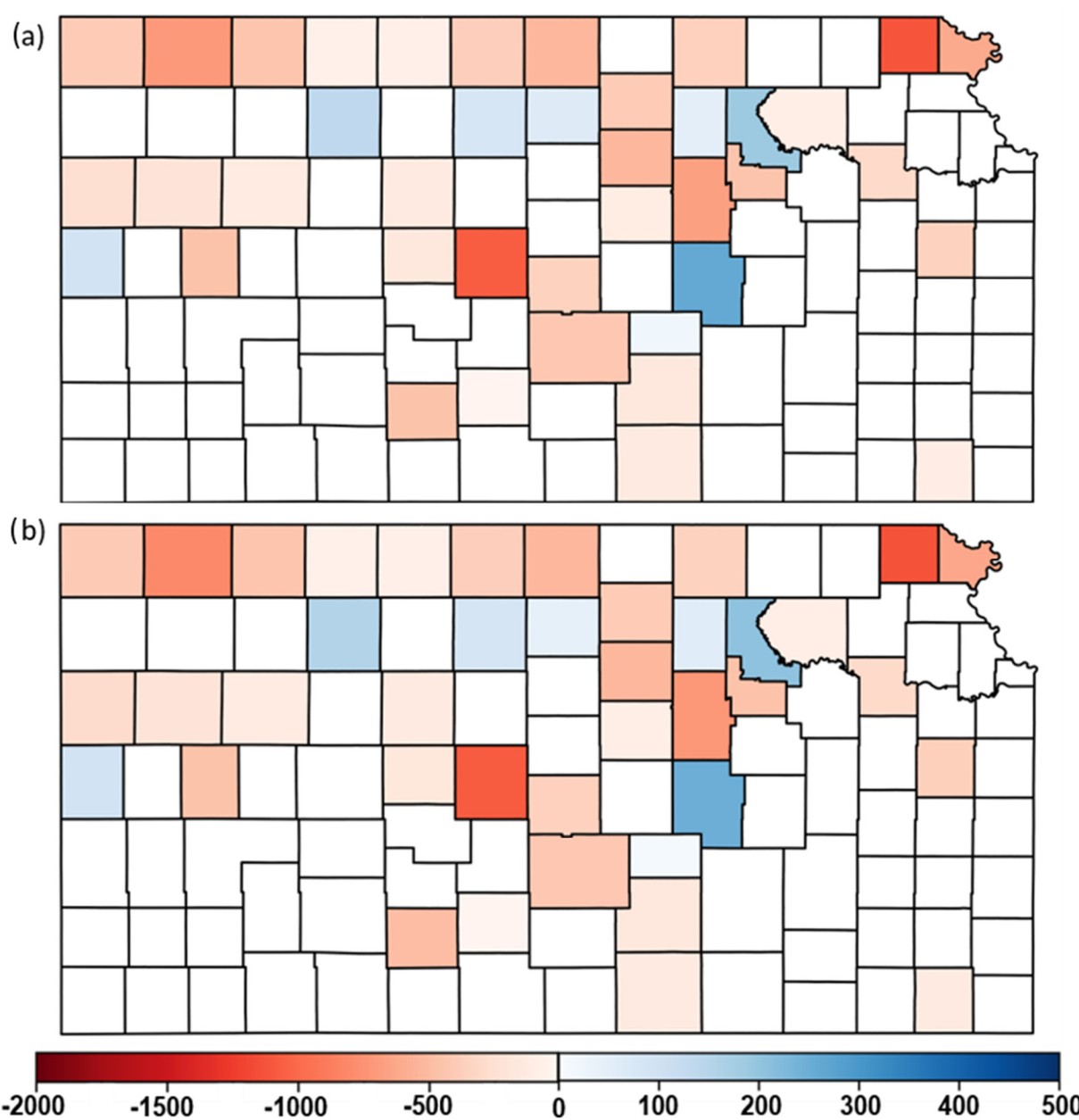

**Figure 7.** Predicted yield changes (kg ha$^{-1}$) for 40 counties of Kansas using a MLR model, for (**a**) Scenario I as +1 °C rise in temperature and (**b**) Scenario II as +2 °C rise in temperature.

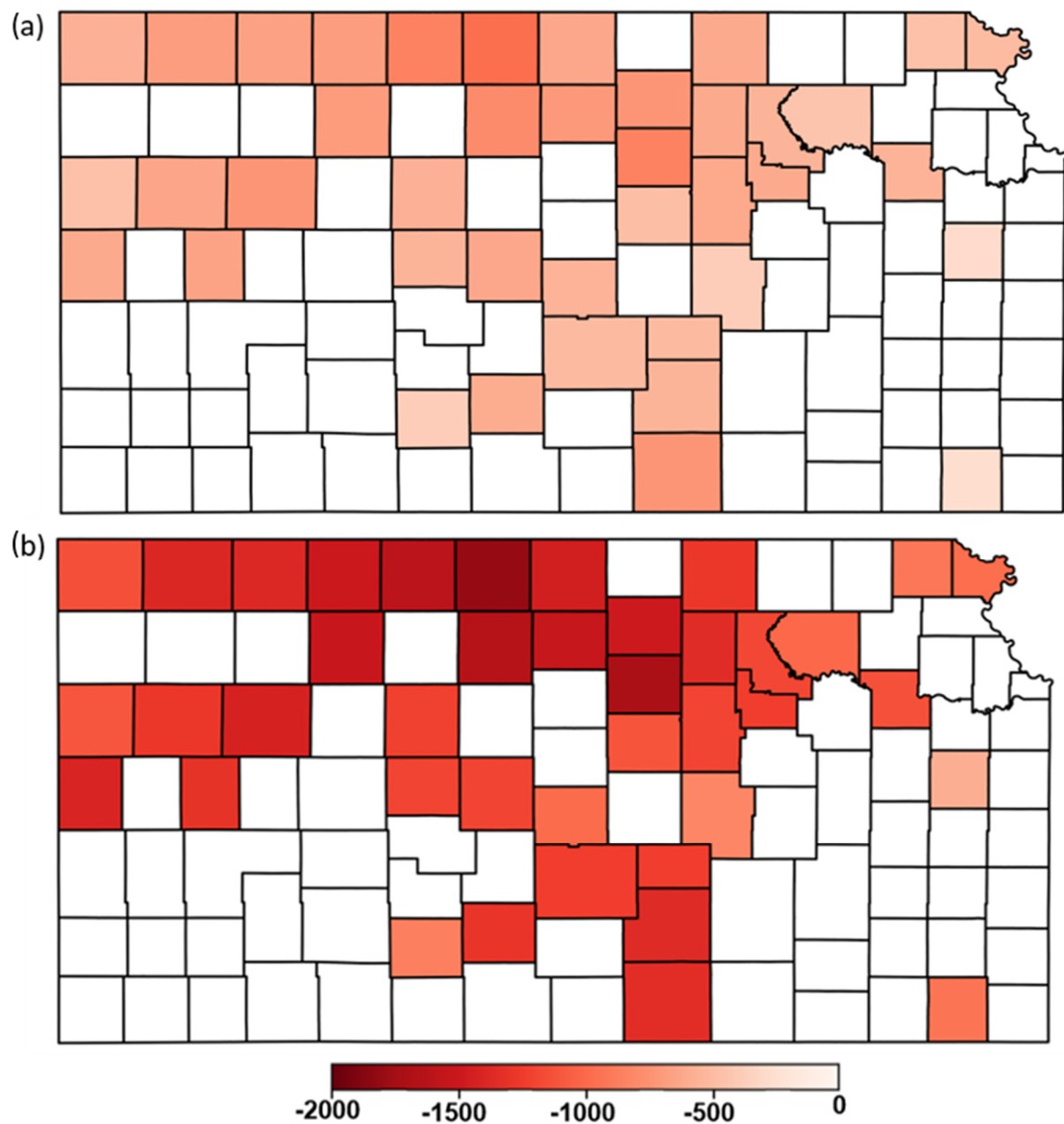

**Figure 8.** Predicted yield changes (kg ha$^{-1}$) for 40 counties of Kansas using the DSSAT model, for (**a**) Scenario I as +1 °C rise in temperature and (**b**) Scenario II as +2 °C rise in temperature.

Overall, the predicted reduction in maize yield was greater in the DSSAT model because the DSSAT model requires daily weather information [44], whereas the MLR model used the average values for weather during the typical growing period of maize [66]. The use of daily weather information in the DSSAT model allows for a more detailed and dynamic representation of crop responses to changing weather conditions, whereas the MLR model's reliance on average weather values for the growing period simplifies the weather input and may not capture the full range of variability experienced with the crop. As a result, daily weather information generally leads to higher sensitivity and responsiveness to weather variations, resulting in greater predicted maize yield reductions for the DSSAT compared to the MLR model. These results are opposite to those found by [29], who assessed the impact of climate change on maize yield and found that predicted impacts are considerably more severe when using the statistical model, combined model

(combination of statistical and process models), and full model (add all interactions with EDD with combined model specification) as compared to the process-based model [10].

Figure 9 displays the overall impact of synthetic climate change scenarios across Kansas. It was found that the reduction in the predicted rainfed maize yield using the MLR model was nearly 6% in both scenarios. These results are similar to those reported from a global maize yield prediction via regression model [24], which found that a 1 °C rise in temperature resulted in an 8.3% yield reduction, as well as those of a study conducted in the central Corn Belt [67] that reported a 5 to 8% reduction in simulated maize yields per 2 °C temperature increase. According to Roberts et al. [29], three different regression models, developed for maize yield prediction in Illinois, found reductions in yield in the range of 10 to 60% for the years 2040–2069, relative to the 1981–2010 baseline scenario. We found reductions in the predicted rainfed maize yield with the DSSAT model of 8.7% and 18.3% for Scenario I and Scenario II, respectively. These results clearly indicate that, irrespective of the type of model used to predict yield, the predicted rainfed maize yield will likely be reduced with an increase in the temperature. The predicted climate change impacts are considerably more severe under the DSSAT process-based model compared to the MLR model, which could be explained by extreme heat effects [36,63], which are not captured with the MLR model because only the DSSAT model uses daily weather information [10].

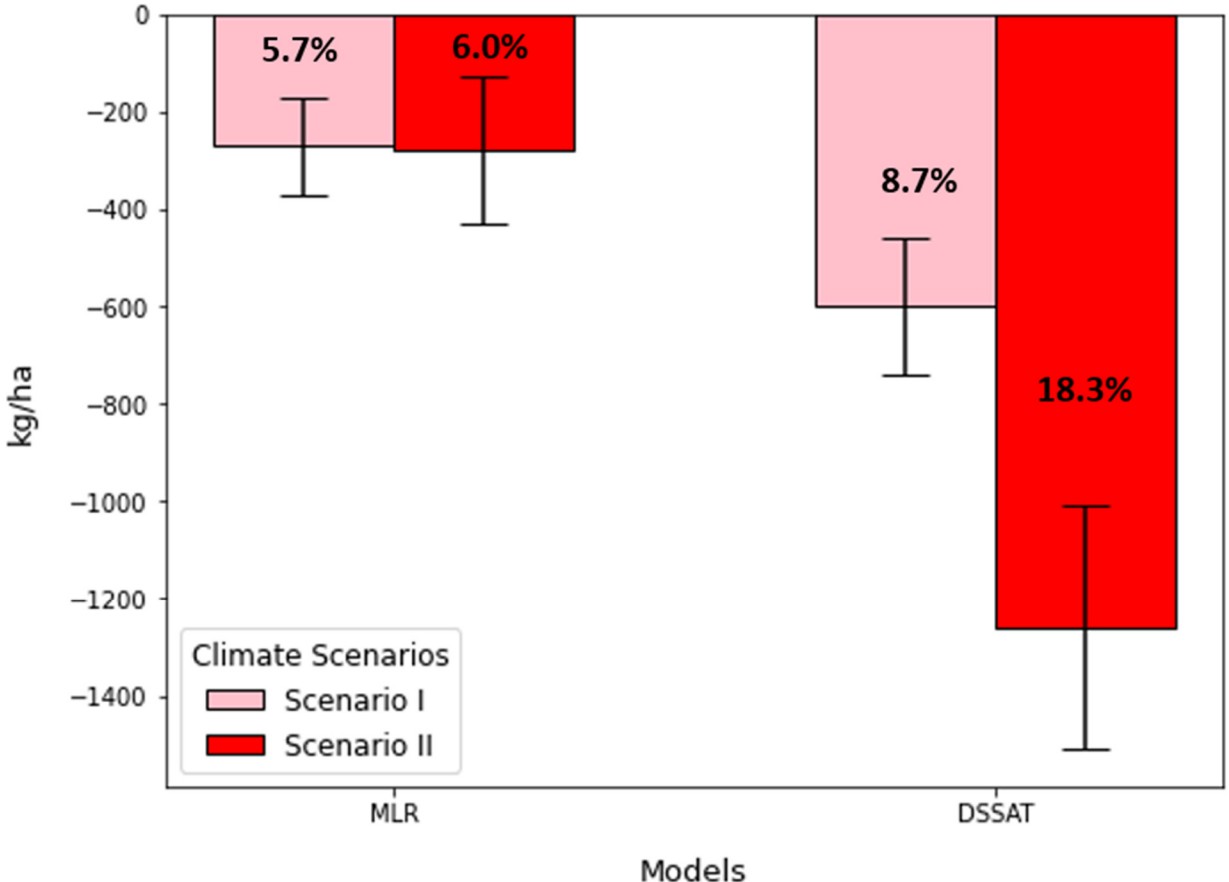

**Figure 9.** Average predicted yield change (kg ha$^{-1}$) for both models and each climate scenario, along with the percentage change in the yield across Kansas [10]. Scenario I is defined as a +1 °C rise and scenario II as a +2 °C rise in daily temperatures for the season.

## 4. Conclusions

In this study, we delved into the understanding of how climate change affects maize yields in the agricultural areas of Kansas. Our research centered around comparing two types of models: a statistical multiple linear regression (MLR) model and a process-based

model (the Decision Support System for Agrotechnology Transfer, DSSAT). By doing so, we aimed to shed light on the most effective approach for predicting maize yields under changing climatic conditions [10].

We found that, using the developed MLR model, at least 32% of the predicted year-to-year rainfed maize yield variability was explained by climate variables in Kansas counties, which suggests that climate plays an important role in rainfed maize yield predictions. We also found that the MLR model outperformed the DSSAT model, as is evident from the higher correlation coefficient (r = 0.93) and lower root mean square error (RMSE = 443 kg ha$^{-1}$) of the MLR model compared to the values of r = 0.70 and RMSE = 1408 kg ha$^{-1}$ that were observed with the DSSAT model. Notably, the DSSAT model's performance hinged largely upon crop management data, such as the planting date and plant population, and soil information, variables that were not a part of the MLR models.

This study demonstrated that rising temperatures generally have a detrimental impact on maize yields in Kansas. As the temperature rose, the predicted reductions in rainfed maize yields were more pronounced, particularly in the DSSAT model. This highlights the vulnerability of maize crops to the intensifying heat that is expected with climate change. However, it is important to recognize the limitations of these models. Both the MLR and the DSSAT models have their respective constraints when it comes to mechanistic understanding, computation demand, and complexity. It is interesting that the two models project very different impacts of climate change. Based on our findings, we think these differences deserve further research, and it might be worthwhile to consider a wider set of models [10,68].

**Supplementary Materials:** The following supporting information can be downloaded at https://www.mdpi.com/article/10.3390/agronomy13102571/s1: Figure S1: Time series of observed (green) and predicted yield from models (simulated yield from DSSAT in red, predicted yield from MLR in blue) where y-axis represents detrended predicted yield (a) Central, (b) North Central, (c) South Central, (d) West Central, (e) North Western, (f) North Eastern, (g) Eastern Central, (h) South Eastern regions in Kansas.

**Author Contributions:** Conceptualization, M.R. and V.S.; methodology, M.R., V.S., X.L. and K.R.; software, M.R. and V.S.; validation, M.R., V.S., X.L. and K.R.; formal analysis, M.R.; investigation, M.R.; resources, M.R.; data curation, M.R.; writing—original draft preparation, M.R.; writing—review and editing, M.R., V.S., X.L. and K.R.; visualization, M.R. and V.S.; supervision V.S., X.L. and K.R. All authors have read and agreed to the published version of the manuscript.

**Funding:** This research received no external funding.

**Data Availability Statement:** Not applicable. All data generated or analyzed during the study are included in this research article. Weather, Soil, and corn yield data are freely available to use from the sites mentioned in this article.

**Acknowledgments:** The author would like to acknowledge the support of the Carl and Melinda Helwig Department of Biological and Agricultural Engineering at Kansas State University. Additionally, a special thanks to Lokendra Singh Rathore, a friend, for their invaluable assistance in data acquisition and coding.

**Conflicts of Interest:** The authors declare no conflict of interest.

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
