# Peer review of "Climate Change Impacts on Rainfed Maize Yields in Kansas: Statistical vs. Process-Based Models"

_agronomy, doi:10.3390/agronomy13102571_

Round 1

Reviewer 1 Report

I suggest improving the Figures because they are unclear regarding abbreviations and explanations for people who do not know the USA.

Add climate data from the study region for better analysis of the results.

I suggest adding more references.

Clarifying why only temperatures were considered when rain is also essential to analyze corn development.

I added a file with specific comments.

Author Response

Please see our response to your comments in the rebuttal document attached. Thank You.

Reviewer 2 Report

This paper investigates the impact of climate change on maize yield in Kansas and draws a comparison between statistical and mechanism-based models. Overall, the paper offers an intriguing comparison, yet the importance and novelty of the study could be better emphasized within the discussion. Furthermore, the elaboration on the results warrants significant enhancement. I am providing the following suggestions for the authors and editors, hoping to further improve the quality of the paper.

Major Comments

1.      The current manuscript exhibits significant overlap with a K-State thesis titled "Comparison of climate change impact on rainfed maize yield in Kansas using statistical and process-based models", especially in the abstract section. I am unaware of the journal's policies regarding the submission of content derived from master's theses. However, I recommend making appropriate revisions to reduce the rate of duplication.

2.      Consider referring to the warming scenarios from IPCC AR6 to choose the conditions for simulated warming.

3.      Statistical models tend to over-simulate when extrapolated beyond the training dataset. Caution is advised when extrapolating with these models, and the study's limitations should be stated in the paper.

4.      Some descriptions are vague, e.g., L17-18.

5.      It's recommended to further explore the significance of variables within the models (sensitivity analysis). Although there's a discussion on important inputs affecting maize growth, the data doesn't clearly show their significance.

6.      Consider overlaying satellite or related imagery to easily identify locations.

7.      Suggest including model parameters of DSSAT and the results of model calibration before large-area application.

8.      Provide scatter plots showing the correlation of simulated results (or place them in supplementary materials).

Minor Comments

1.      The term "recent" may be interpreted differently by various readers. Consider specifying the timeframe of analysis in the title.

2.      When mentioning simulation data in the abstract, it would be helpful to indicate the context and scope of the predictions.

3.      In the introduction, while maize's significance is mentioned, the focus might be better placed strictly on the research topic. The current introduction seems a bit broad, discussing other crops like wheat and rice.

4.      The introduction states that mechanism models lack extreme data, leading to potential inaccuracies during simulations. This also applies to statistical models.

5.      The definition of climate zones is unclear. Are the authors referring to zones b and c under the Köppen climate classification system? A brief explanation would be helpful.

6.      Suggest presenting Figure 1 in a conventional map format, including a north arrow, latitude/longitude, legend, and scale.

7.      Consider reordering the sections. You've quickly introduced data sources in the Materials and Methods section, but readers might not yet understand the parameters used for model development. Begin with the study area's characteristics, followed by the model and its parameters, and then introduce data sources.

8.      Ensure the completeness of Equation 1. For instance, if Tx,d= Tlow but > Tbaseline, what would Tx,d be?

9.      For all equations, define each variable. For example, Equation 1 lacks Thigh and Tlow. Additionally, consider reformatting the equations for better readability.

10.  Please provide details on the maize varieties, types of fertilizers, and amounts of fertilizers used in each county.

11.  Justify the use of only one maize variety for simulating maize yield across the state with DSSAT, and provide a list detailing the input items and variety parameters used in the DSSAT model.

12.  Please specify the data volume, data preprocessing measures and methods, and the rationale for selecting 6 out of 9 variables (why not 3, 8, or others?).

13.  Consider illustrating the distribution of the dataset, like a box plot.

14.  In the description for Equation 4, the authors mentioned the average of predicted and observed values, but these aren't used in Equation 4. Please relocate to the appropriate section.

15.  What is the function of 'f' in Equation 6?

16.  In Figure 2 on page 7, it's unclear what you're trying to convey. Are the authors suggesting that the lower determination coefficient in some regions is due to topographical or climatic factors? If you're only highlighting the relationship between the determination coefficient and climate, this figure might not be necessary. Also, numerous white areas hinder interpretation; consider revising the presentation method.

17.  Similar comments apply to Figures 3 and 4.

18.  On pages 9-10, the authors compared the simulation results for eastern and western Kansas and found no difference. However, the authors didn't specify how the authors distinguished between the east and west, nor why such a comparison was made.

19.  In Figure 5 on page 10, why are there no values for the DSSAT model around 1981-1983?

20.  The statement on page 11, L339-342, seems vague.

21.  On page 13, the authors discuss the difference in simulation outcomes for the two models under two warming scenarios. However, some key arguments require additional citations for support.

22.  The y-axis label in Figure 8 should be consistent with previous figures.

23.  I'm uncertain about the comparability of detrended yield with results from previous studies. Could you clarify?

24.  Minor adjustments: Define acronyms the first time they appear (e.g., APSIM, DSSAT). Use italics for scientific names. Avoid excessive citations where a single reference would suffice. Adjust text formatting issues (e.g., missing space after a period in L189). Improve the resolution of figures where needed (e.g., Figure 8).

English writing is fine.

Author Response

(The authors gave the same response as above.)

Round 2

Reviewer 1 Report

I made a few comments in the manuscript.

It will be easier to review the the this document in Word rather than in PDF.
